# Impact of Health and Social Factors on the Cardiometabolic Risk in People with Food Insecurity: A Systematic Review

**DOI:** 10.3390/ijerph192114447

**Published:** 2022-11-04

**Authors:** Blanca Salinas-Roca, Laura Rubió-Piqué, Elena Carrillo-Álvarez, Gemma Franco-Alcaine

**Affiliations:** 1Department of Nursing and Physiotherapy, University of Lleida, Montserrat Roig 2, 25198 Lleida, Spain; 2Global Research on Wellbeing (GRoW) Research Group, Blanquerna School of Health Science, Ramon Llull University, Padilla, 326-332, 08025 Barcelona, Spain; 3Antioxidants Research Group, Food Technology Department, AGROTECNIO-CERCA Center, University of Lleida, Av/Alcalde Rovira Roure 191, 25198 Lleida, Spain

**Keywords:** food insecurity, cardiometabolic, non communicable diseases, eating behavior, health-determinants, social-determinants

## Abstract

Food plays a key role in people’s health and quality of life. Inadequate eating habits or a deficient diet can lead to the development of non-communicable diseases (NCDs). The present review aims to describe the health and social factors related to food insecurity (FI) in adults in high-income countries and evaluate their impact on cardiometabolic risk (CMR). Following the PRISMA procedures, a systematic review was conducted by searching in biomedical databases. Full articles were screened (nf = 228) and critically appraised, and 12 studies met the inclusion criteria. Based on the selected studies, the results grouped information based on (i) the characteristics of the population in FI, (ii) the impact of FI on NCDs, and (iii) the cardiovascular and all-cause mortality risk of the FI population. Considering the minimum and maximum percentage data, people of the categories female sex (46.2–57.6%), education level lower than high school (11–67.46%), non-Hispanic white ethnicity (37.4–58%), single or separated or widowed (45–64.8%), and current smoker (35.5–61.1%) make up the population with FI in high-income countries. All of these factors presented a significant association (*p* < 0.001) with cardiovascular risk factors. The highest odds ratios (OR) for the FI population are described for obesity (OR = 2.49, 95% CI; 1.16–5.33) and myocardial infarction (OR = 2.19, 95% CI). Interventions that integrate FI screening and the measurement of CMR factors into routine clinical care may be an important step to identify vulnerable populations and subsequently improve and prevent NCDs. Thus, food-diet policies and public-health-based interventions are needed to be included in the measurement of CMR in the assessment of FI.

## 1. Introduction

The Food and Agriculture Organization of the United Nations (FAO) defines food insecurity (FI) as insufficient food intake, which can be transient (when it occurs in times of crisis), seasonal (when it is linked with the weather seasonality), or chronic (when it happens continuously). It can also be classified as mild, moderate, or severe, depending on the level of access to food [1,2].

Access to a sufficient amount of food is as important as access to a variety of food, which is necessary for a balanced and healthy diet. Food plays a key role in people’s health and quality of life. Inadequate nutritional habits or a deficient diet can lead to metabolic disorders and consequently the development of chronic non-communicable diseases (NCDs), such as cardiovascular disease (CVD) and diabetes mellitus (DM) [3,4,5,6,7].

A state of FI can lead to the appearance of compensatory behaviors, such as cycles of abundance and deficiency, the omission of meals, and the ingestion of foods with little nutritional value [1,8,9]. These behaviors worsen the quality of the diet and can alter physiological functions, leading to increased blood glucose; insulin resistance; arteriosclerosis; and increased blood pressure, among others, which are risk factors for NCDs, which together account for almost 70% of the world’s deaths [10,11,12]. Most of these deaths could be prevented with the early detection of NCDs, especially those most closely linked to the cardiometabolic level, such as hypertension and diabetes, and lifestyle, such as physical activity or toxic habits [13,14,15].

The close relationship between FI and the emergence of NCD has been demonstrated [12,16,17,18]. However, it is necessary to investigate the risk factors present in the food-insecure population, which are most associated with NCD morbidity and mortality, in order to be able to detect those most vulnerable people at an early stage.

In recent decades, numerous studies have been conducted on cardiovascular risk estimation such as Framingham risk scores from the Framingham Heart Study (FHS) [19,20], QRISK equations [21], Europe risk equations SCORE [22], Scottish Heart Health Extended Cohort (SHHEC) ASSIGN scores [23], Prospective Cardiovascular Master (PROCAM) equations [24], and CUORE Cohort Study formulas [25]. These CVD risk-prediction models have been shown to be effective in early health and disease management for physicians and individuals [26,27,28]. However, none of them are specific to people with FI or at risk of suffering from it.

Much work is currently being done worldwide to guarantee the right to access food in a dignified manner, and food is increasingly taking a position as a basis for life, a determinant of health, and an empowering element for all people [1]. Although the end of hunger and the improvement of health are included in the seventeen sustainable development goals (SDGs) set by the UN for 2030, especially SDGs 2 and 3, indicators of food poverty are worsening. Indeed, people suffering from FI have increased to more than 30% of the world’s population after the COVID-19 pandemic [1,2,29].

Therefore, it is necessary to address this paradigm with the assistance of the FI and study the current health impact in order to understand the magnitude of FI, make it visible, and encourage the creation of food policies for reversing the situation. Thus, the present review aims to describe the health and social factors related to FI in adults in high-income countries and evaluate their impact on cardiometabolic risk (CMR).

## 2. Materials and Methods

The guiding question employed in this research was “which social and health elements are identified as risk factors for cardiometabolic disease in the FI population?”. The question is based on the rising incidence of FI worldwide and the non-communicable diseases related to cardiometabolic risk.

### 2.1. Protocol and Criteria for Selection of Articles

This systematic review was conducted according to the PRISMA protocol-Preferred Reporting Items for Systematic Reviews [30]. All excluded studies were those not fitting the inclusion criteria in Table 1.

### 2.2. Search, Selection and Evaluation of Articles

In order to identify the health and social risk factors of suffering NCDs related to FI, an exhaustive search was conducted in the PubMed and Scopus databases using the PRISMA methodology [30]. To carry out the search, a combination of keywords and Boolean operators (such as AND, OR, NOT) were included, and articles were filtered according to inclusion and exclusion criteria. The search was carried out between October 2021 and April 2022. Thus, the search strategy considers the following search terms: (food insecurity) AND (cardiometabolic) NOT (low-income countries) NOT (infant population); (food insecurity) AND (cardiometabolic) AND (poverty) NOT (low-income countries) NOT (infant population); (food insecurity) AND (cardiovascular)) AND (poverty) AND (mortality) NOT (low-income countries) NOT (infant population) and (food insecurity) AND (cardiovascular) AND (mortality).

Once the search strategies were carried out and the filters were applied according to the inclusion and exclusion criteria, 57 articles were identified. Title and abstract screening resulted in the exclusion of 31 additional papers. The resulting 26 articles were read in full text to identify whether they met the inclusion criteria. In this last phase, 14 articles were excluded, and finally 12 articles were included to develop the bibliographic review. See Figure 1 for the complete flowchart.

The evaluation of the data in the selected articles was performed by analyzing the type of study, the country, the population, and the association between FI and CMR. Thus, adjusted HR from the original articles selected in the review were included in the section of results.

## 3. Results

After the exhaustive literature search, the screening of articles, and their detailed evaluation, 12 articles were included in the study (Figure 1). Five studies examine the associations between FI and overall mortality [17,18,31,32,33], from CVD [17], from chronic diseases such as cardiorenal syndrome [31], and from premature mortality [32]. Five studies determine the association between FI and NCDs [34,35,36,37], including DM, hypertension (HT), obesity, and chronic obstructive pulmonary disease [34]. Finally, two studies investigate the association between FI and the risk of suffering CVD [38,39].

Of the 12 studies, ten are from the United States [17,18,31,33,34,36,37,38,39,40], one is from Canada [32], and one is from The Netherlands [35]. The ten American and Canadian studies are retrospective cohort studies with an adult population, drawing data from the National Health and Nutrition Examination Survey (NANHES) [17,18,31,33,34], the National Health Interview Survey (NHIS) [40], the National Longitudinal Study of Adolescent to Adult Health [36], a regional baseline survey, the Mississippi Behavioral Risk Factor Surveillance System [38], and the Community Health Survey Canadian [32]. The Dutch study is a cross-sectional mediation analysis with an adult population and with the collection of data using questionnaires [35].

The methodological quality of the included studies was assessed using the adapted CASPe guide, where all studies presented a total score between 7 and 9 out of 9 [41].

Based on the selected studies, the results grouped information based on the (i) characteristics of the population in FI (ii) impact of FI on NCDs and (iii) mortality and cardiometabolic risk of FI population. Table 2 shows the socioeconomic and health characteristics of the FI population (represented in percentages) that presented a significant association with cardiovascular risk (*p* value < 0.001). All of the authors agree on the description of females as the gender influenced the most by FI [17,33,36,38,39], and education below high school [17,18,32,34,37,38,39], showing a compelling association with cardiovascular risk. Among toxic habits, smoking has a significant effect on NCDs in adults suffering FI [17,18,31,33,36,38,39]. Table 3 describes the influence of FI on the appearance of NCDs represented by adjusted odds ratios (OR) and 95% confidence interval (CI). The greatest evidence exists for HT (4 studies, OR 1.40–1.51), obesity (4 studies, OR 1.60–2.49), and DM (4 studies, OR 1.23–1.67). Other outcomes identified as cardiovascular risk factors include myocardial infarction, coronary heart disease, angina pectoris, stroke, and smoking. Among the 12 studies, 2 evaluated the mortality from CVD in the population with FI. Table 4 shows these data as hazard ratios (HR) and 95% CI. A significant relation between mortality and male population under FI and DM was observed.

## 4. Discussion

In this review, we sought to describe the health and social factors related to FI in adults in high-income countries and evaluate their impact on cardiometabolic risk. Our results show a close relationship between FI and risk factors for suffering from diet-related NCDs [17,18]. In that sense, some studies also described high mortality from all causes and from CVD in people with FI [32,33].

According to the results presented in Table 2, the individual characteristics more prevalent within the population suffering from FI are female sex, lower educational level, white non-Hispanic ethnicity, single, separated or widowed, and current smoker.

The prevalence of the female sex over the male can be explained by gender inequalities in fundamental areas that affect access to food, such as the participation in the labor market or the distribution of income within the household, which shows a strong relationship with FI in all countries [42]. Several mechanisms have been described to explain the association of educational level, and eating habits. Higher educational attainment not only brings more food literacy but also makes people value health more and produces higher self-efficacy, as well as less correct selection, consumption, use, storage, and hygienic conditions related to food. This poor eating pattern will affect the whole family [43]. Therefore, if we know that women contribute significantly to the feeding of households, we can say that the level of food security of children depends on the education of their mothers, and their access to social and economic resources.

The relationship between the higher prevalence of FI in single, separated, or widowed people refers mainly to people living alone [7,8]. The eating habits of people who live alone tend to be more flexible and with a greater presence of compensatory behaviors that are not socially accepted, such as skipping meals, substituting them for coffee or cigarettes, or “bingeing” ultra-processed cheap and comfort food with high. Additionally, the quality of the preparation of the meals tends to be lower as they have to be prepared by a single person and tend to cook more simply or based on preparations [44].

Another individual characteristic in people suffering FI with a relation with NCDs is smoking habit. Smoking has also been associated with FI; this fact can be attributed to the proven satiating power that tobacco causes. A study published reveals how nicotine activates a group of neurons in the hypothalamus that transmit the feeling of satiety [45]. This can lead to smokers reducing their intake and even replacing meals with cigarettes. A further factor that can explain this association is the fact that addiction to tobacco drives these people to use the money they could spend on buying food, on buying packs of cigarettes.

The characteristics directly associated with increased mortality differ from those that are more prevalent within the FI population. According to recent evidence, mortality associated with FI is increased in those who are male; less than high school educated; non-Hispanic black; single; separated or widowed; current smokers; obese (BMI > 30); and with the presence of DM, HT, CVD or chronic kidney disease [17,18,31,32,33].

Additionally, compensatory eating behaviors can explain this increase in morbidity and mortality associated with FI. To describe the physiological implications of these behaviors and which diseases they can develop, Figure 2 shows the deterioration of health caused by the lack of access to food, based on the recent evidence that relates these terms [43,44,45,46,47]. The reviewed literature indicates that the main diet-related metabolic risk factors associated with the development of CVD and DM are overweight/obesity (OO), hyperglycemia, elevated blood lipids, and HT [21].

The present review describes the influence of FI on the occurrence of NCDs related at a cardiometabolic level, such as coronary heart disease [34,40], myocardial infarction [34,40], angina pectoris [40], stroke [34], HT [34,36,37,38], DM [34,36,37,38], and obesity [35,37,38,40], as well as toxic habits such as smoking [39]. Specifically, the highest OR are described for obesity and myocardial infarction. This could be due in part to the poor eating habits as a consequence of FI states, with the intake of cheap ultra-processed foods loaded with fast-absorbing carbohydrates, which increase blood glucose, and saturated fats, which cause arteriosclerosis [13,14,15]. In accordance, Figure 2 shows how FI drives food-related or lifestyle compensatory behaviors. Those compensatory behaviors are linked to mental disorders and have an outstanding relation to cardiometabolic-risk diseases.

People living in FI often experience an abundance-scarcity cycle, where food intake fluctuates depending on food availability, so food intake decreases during periods of food scarcity and increases during periods of relative food abundance, for example, after receiving food aid. This cycle can be amplified by some approaches used to mitigate FI that are not well managed [8,44]. For example, food-distribution programs, which tend to provide benefits once a month, and are often insufficient [46,47,48]. These benefits are usually consumed disproportionately soon after receiving them and are used up before the end of the month, which causes food restriction with the respective risk of binge eating. These behaviors cause an excessive intake of carbohydrates over a period of time that can lead to eating disorders, obesity, and DM, among others [8,44,49].

This excessive consumption of foods with a high calorie content can also be explained by their low cost. Families in vulnerable situations are driven to buy food with less nutritive value such as high sugar and salt food and ultra-processed food [8,9]. This type of food based on carbohydrates and saturated fats has a low nutritional value, and their consumption as the basis of the diet can cause a serious deficit of micronutrients, proteins, and minerals, which are essential to maintain a good state of health [44,50]. This prolonged feeding over time can lead to an increase in triglycerides, cholesterol, blood glucose, iron, calcium, vitamins, and fiber deficiency, among others, and end up developing DM, CVD, or anemia, among many other conditions [50,51].

In agreement with the literature, special attention should be paid to hypertension due to the food-related compensatory behaviors of the FI population as well as the easily underdiagnosed pathology [52]. According to prior research, a greater number of socially determined vulnerabilities were associated with progressively higher risk of developing hypertension, and an even higher risk of dying over 10 years [5]. In that sense, established risk factors include an unhealthy diet (high salt and low fruit and vegetable intake), physical inactivity, tobacco and alcohol use, and obesity. Emerging risk factors include pollution (air, water, noise, and light), urbanization, and a loss of green space. Risk factors that require further in-depth research include the social and commercial determinants of health [53].

## 5. Conclusions

This review provided a step forward in understanding the association between FI and the increased cardiometabolic risk associated with NCDs, especially conveyed through a higher prevalence of obesity and myocardial infarction. Existing evidence confirms FI as a risk factor for suffering from diet-related NCDs. The population with FI in high-income countries is represented in its majority by people of the female sex, who have a level of education less than high school; are of white non-Hispanic ethnicity; are single, separated, or widowed; and are current smokers. Individuals with this profile are more likely to have FI and more likely to suffer from NCDs linked with cardiometabolic risk.

On the other hand, mortality from NCDs associated with FI increases in a vulnerable profile that is slightly different from the previous one: male sex, who have a level of education less than secondary; are of non-Hispanic black ethnicity; are single, separated, or widowed; are current smokers; are obese (BMI > 30); and have of DM, HT, or CVD. Therefore, due to its association with mortality, these are the characteristics that must be kept in mind in order to achieve the optimal early detection of cardiometabolic risk in people with FI.

Thus, it is necessary to include the measurement of CVD risk factors such as hypertension in the assessment of the level of FI and in the prevention of NCDs. The present review summarizes the current comprehensive epidemiological evidence that may inform specific intervention strategies targeting food-insecure groups to reduce CVD risk. Future research and public health-based strategies are urgently needed to recognize and guarantee the right to food to the most vulnerable groups, which includes decent access to a sufficient quality dietary pattern.

## Figures and Tables

**Figure 1 ijerph-19-14447-f001:**
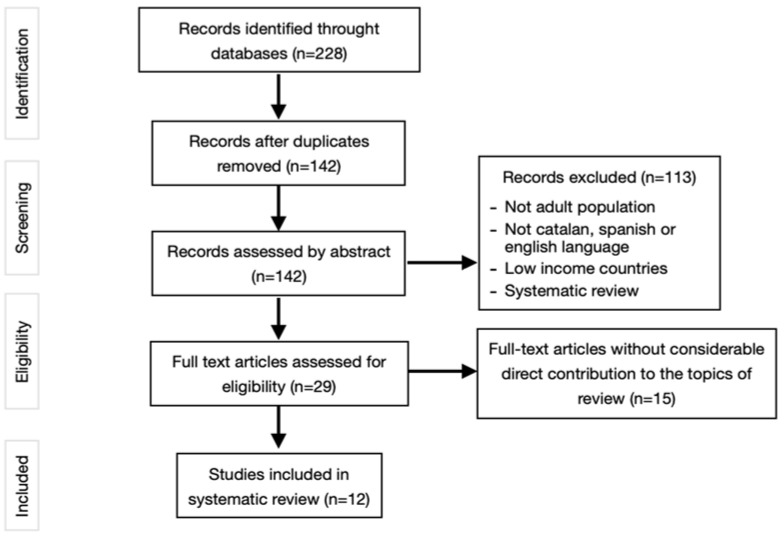
Flow chart showing the electronic research, identification, and selection process for the reviewed articles.

**Figure 2 ijerph-19-14447-f002:**
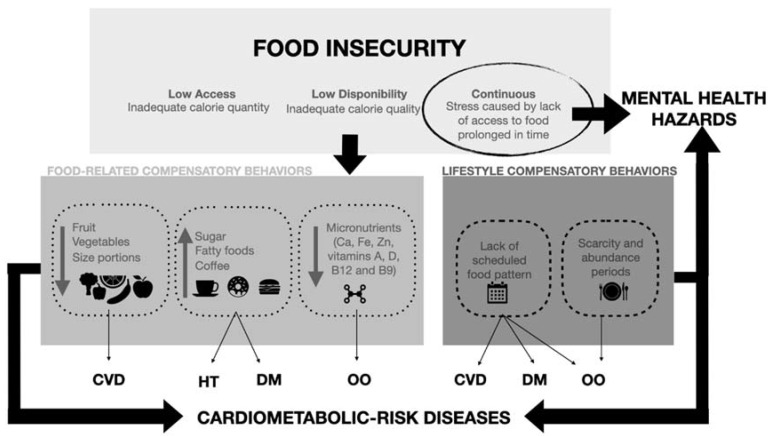
Conceptual framework showing the relationship between eating behavior and cardiometabolic risk-related diseases and mental wellness in food insecurity context. Abbreviations; CVD: cardiovascular disease, HT: hypertension, DM: diabetes mellitus, and OO: overweight and obesity.

**Table 1 ijerph-19-14447-t001:** Inclusion criteria considered for the selection of articles.

Inclusion Criteria	
Studies published after 2016	Adult population
Studies in Catalan, Spanish or English	No pregnant women
High income and middle-income countries	Cohort study; a randomized, controlled trial; and cross-sectional studies

**Table 2 ijerph-19-14447-t002:** Percentage of population under food insecurity, according to several socioeconomic and health characteristics that presented a significant association with cardiometabolic risk (all data are *p* < 0.001, if not detailed).

	Sun et al. (2020) [17]	Banerjee et al. (2021) [18]	Radak et al. (2019) [31]	Nagata et al. (2019) [36]	Helmick et al. (2018) [37]	Mendy et al. (2018) [38]	Verccammen et al. (2019) [39]	Walker et al. (2019) [33]
**Female**	55.6%	-	-	57.6%	-	46.2% (*p* = 0.0011)	52.7% (*p* = 0.06)	53.52% (*p* = 0.04)
**Education: less than high school**	42.7%	40.1%	42.55%	18.6%	11%	57.8%	-	67.46%
**Ethnicity: Non-hispanic White**	-	-	37.4%	58%	-	36.4%	51.8%	45.41%
**Ethnicity: Non-hispanic black**	-	-	17%	25.3%	-	53.7%	18.6%	19.97%
**Ethnicity: ** **hispanic**	-	-	29%	10.6%	-	-	23.4%	28.48%
**Married/domestic ** **partnership**	-	-	-	-	55%	35.2%	47.3%	38.72%
**Not married**	-	-	-	-	45%	64.8%	52.7%	61.22%
**Current Smoking**	44.3% (2.3)	37.8%	35.5%	46.3%	-	61.1%	46.3%	40.61%
**Obesity**	-	-	32.97%	43.3%	68% (*p* = 0.003)	49.5		38.59%

**Table 3 ijerph-19-14447-t003:** Influence of food insecurity in the development of NCDs, expressed as odds ratio and 95% CI.

	Venci et al. (2018) [34]	Van der Velde et al. (2020) [35]	Nagata et al. (2019) [36]	Helmick et al. (2018) [37]	Smith et al. (2019) [40]	Mendy et al. (2018) [38]	Verccammen et al. (2019) [39]
**Coronary heart disease**	1.75 (1.37–2.24) ns	-	-	-	1.56 (1.85 in female) (*p* < 0.001)	-	-
**Myocardial infarction**	1.40 (1.08–1.81) ns	-	-	-	2.19 (4.04 in female) (*p* < 0.001)	-	-
**Angina pectoris**	-	-	-	-	1.81 (2.5 in female) (*p* < 0.001)	-	-
**Stroke**	1.10 (0.82–1.48) ns	-	-	-	-	-	-
**Hypertension**	1.42 (1.22–1.65) ns	-	1.40 (1.14–1.72) (*p* = 0.002)	1.45 (1.11–1.90) (*p* = 0.004)	-	1.51 (1.21–1.88) (*p* < 0.001)	-
**Diabetes Mellitus**	1.23 (1.02–1.48) ns	-	1.67 (1.18–2.37) (*p* = 0.004)	1.52 (1.09–2.13) (*p* = 0.014)	-	1.30 (1.02–1.65) (*p* = 0.0365)	-
**Obesity**	-	2.49 (1.16–5.33) (*p* < 0.02)	-	1.60 (1.17–2.20) (*p* = 0.004)	1.78 (1.34 in female) (*p* < 0.001)	1.68 (1.28–2.21) (*p* < 0.001)	-
**Current Smoking**	-	-	-	-	-	-	1.95 (1.60–2.37) (*p* < 0.001)

ns: non-significant.

**Table 4 ijerph-19-14447-t004:** Cardiovascular and all-cause mortality risk in food insecurity population, expressed as Hazard ratio (HR) and 95% CI. All data are *p* < 0.001, if not detailed.

		Banerjee et al. (2021) [18]	Walker et al. (2019) [33]
All-cause mortality in FI population (adjusted HRs)	General	1.46 (1.23–1.72)	-
	Male	1.62 (1.48–1.78)	-
	Education: less than high school	1.48 (1.30–1.68)	-
	Not married	-	1.87 (1.54–2.26)
	Current Smoking	-	2.00 (1.62–2.48)
	Diabetes Mellitus	1.70 (1.47–1.97)	1.42 (1.21–1.67)
CVD mortality in FI population (adjusted HRs)	General	1.75 (1.19–2.57) (*p* < 0.01)	-
	Male	1.99 (1.56–2.53)	-
	Ethnicity: Non-hispanic black	1.54 (1.14–2.09) (*p* < 0.01)	-
	Diabetes Mellitus	1.44 (1.09–1.90) (*p* < 0.05)	-

## Data Availability

Available upon request to corresponding author (blanca.salinasroca@udl.cat).

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
