# Peer review of "Impact of Health and Social Factors on the Cardiometabolic Risk in People with Food Insecurity: A Systematic Review"

_ijerph, 2022, doi:10.3390/ijerph192114447_

Round 1

Reviewer 1 Report

Lines 18-21: It’s not clear what the meaning of the % ranges is. Are those the min-max % from the papers included? Please clarify.

Line 22: Typo (missing a point between sentences). 

Lines 22-23: I strongly suggest providing the 95% Cis for the odds ratios instead of the p-values, or both if space allows it.

Lines 23-26: The conclusion of the review is not fully supported by the data from this paper.

Line 82: Table 1 indicates that one of the inclusion criteria for this systematic review is cohort studies and RCTs. However, the authors included some cross-sectional studies like NHANES and the Dutch study. All cross-sectional studies should be excluded from the systematic review, or the inclusion criteria should be changed. Also, it’s not clear the rationale for conducting the search for papers published after 2016. 

Lines 83-102: In Table 4, the authors included the adjusted HR. In the Materials and Methods section, I recommend indicating how the adjusted HRs were selected from the original papers (e.g., the fully adjusted model? Excluded models that were adjusted for intermediaries?)

Lines 123-124: How was the methodological quality conducted? I strongly recommend by duplicate (by 2 authors) and authors should clarify how discrepancies between authors are handled. 

Author Response

Thank you for your comments. We appreciate the time and effort that you have dedicated to providing your feedback. We believe that the reviewer comments have identified important issues which required improvement, and that after completion of the suggested edits the revised manuscript has benefitted from an improvement. Please, find enclosed the new version of the manuscript with changes marked up using the “Track Changes” function.

REVIEWER 1

Lines 18-21: It’s not clear what the meaning of the % ranges is. Are those the min-max % from the papers included? Please clarify.

In lines 18-21 the percentages described the range (min-max%) of population suffering food insecurity. Data is obtained from the papers included. As reviewer 1 suggested is clarified in the paper as:

Line 22: Typo (missing a point between sentences). 

As suggested by the reviewer, authors have solved the missing point.

Lines 22-23: I strongly suggest providing the 95% Cis for the odds ratios instead of the p-values, or both if space allows it.

Following the suggestion of the reviewer, the 95% CI has been added in the revised text and completed as following: Considering the minimum and maximum percentage data, people of female sex (46.2-57.6%), education level lower than high school (11-67.46%), non-Hispanic white ethnicity (37.4-58%), single or separated or widowed (45-64.8%), and current smoker (35.5-61.1%) represent the population with FI in high-income countries.

Furthermore, and following your suggestion the information regarding the 95 % Cis for the OR has been added in the tables of the results and also detailed explain in lines 136-144.

Lines 23-26: The conclusion of the review is not fully supported by the data from this paper.

We appreciate your suggestion therefore the conclusion has been changed to:

Interventions that integrate FI screening and the measurement of CMR factors into routine clinical care, may be an important step to identify vulnerable population and subsequently improve and prevent NCDs. Thus, food-diet policies and public health-based interventions are needed to be included in the measurement of CMR in assessment of FI.

Line 82: Table 1 indicates that one of the inclusion criteria for this systematic review is cohort studies and RCTs. However, the authors included some cross-sectional studies like NHANES and the Dutch study. All cross-sectional studies should be excluded from the systematic review, or the inclusion criteria should be changed. Also, it’s not clear the rationale for conducting the search for papers published after 2016. 

As requested by the reviewer, this information has been added to the inclusion criteria. Furthermore, the articles selected have been delimitated by 2016 to 2022 since Sustainable Development Goals appeared in that year.

Lines 83-102: In Table 4, the authors included the adjusted HR. In the Materials and Methods section, I recommend indicating how the adjusted HRs were selected from the original papers (e.g., the fully adjusted model? Excluded models that were adjusted for intermediaries?)

Considering the reviewer suggestion, additional information has been detailed in Materials and Methods. Therefore, in line 102-103 the following sentence was added: Thus, fully? adjusted HR from the original articles selected in the review were included in the section of results

Lines 123-124: How was the methodological quality conducted? I strongly recommend by duplicate (by 2 authors) and authors should clarify how discrepancies between authors are handled. 

As the reviewer 1 commented the selection of articles and the quality evaluation was performed in duplicate and discrepancies were handled by a third author disparity.

Reviewer 2 Report

Impact of health and social factors on the cardiometabolic risk 2

in people with food insecurity: a systematic review

1.0 Introduction

Line 38-40 Please explain further how the inadequate diet influence the development NCD

Line 66-67 please rephrase the sentence, Moreover, and after the Covid-19 pandemic the number of people suffering from FI has increased to more than 30% of the world's population

Line 49-50 please explain further

Materials and methods

Materials and methods

Line 89-94: Typing error)))

Line 96-97: No need to mention duplicates were removed. Please directly mention 57 articles were identified

Results and discussion

It is suggested to add the table of nutrient contents of low-cost food that is normally purchased by the FI group and discussed further

Thank you

Author Response

Thank you for your comments. We appreciate the time and effort that you have dedicated to providing your feedback. We believe that the reviewer comments have identified important issues which required improvement, and that after completion of the suggested edits the revised manuscript has benefitted from an improvement. Please, find enclosed the new version of the manuscript with changes marked up using the “Track Changes” function.

R2

Introduction

Line 38-40 Please explain further how the inadequate diet influence the development NCD

As reviewer suggested additional information on metabolic disorders by non-adequate dietary habits haave been mentioned as following: Food plays a key role in people's health and quality of life. Inadequate nutritional habits or a deficient diet can lead to metabolic disorders and consequently

Line 66-67 please rephrase the sentence, Moreover, and after the Covid-19 pandemic the number of people suffering from FI has increased to more than 30% of the world's population

The sentence has been modified to Indeed, people suffering from FI has increased to more than 30% of the world's population after the Covid-19 pandemic

Line 49-50 please explain further

The information has been specified so that sentence changed to Most of these deaths could be prevented with early detection of NCDs, especially those most closely linked to the cardiometabolic level as hypertension and diabetes, and lifestyle such as physical activity or toxic habits.

Materials and methods

Line 89-94: Typing error)))

As reviewer 2 suggested the typing error from lines 89-104 have been solved.

Line 96-97: No need to mention duplicates were removed. Please directly mention 57 articles were identified

Following the suggestions sentence have been reformulated.

Results and discussion

It is suggested to add the table of nutrient contents of low-cost food that is normally purchased by the FI group and discussed further

Since no all papers are contributing to food dietary data authors considered better not mentioned as a table but mentioned in the text that not enough data have been collected and further research on food low cost would be needed.

Thank you
